



# Brief Communication: New radar constraints support presence of ice older than 1.5 Ma at Little Dome C

David A. Lilien[1,*], Daniel Steinhage[2,*], Drew Taylor[3], Frédéric Parrenin[4], Catherine Ritz[4], Robert Mulvaney[5], Carlos Martín[5], Jie-Bang Yan[3], Charles O'Neill[3], Massimo Frezzotti[6], Heinrich Miller[2], Prasad Gogineni[3], Dorthe Dahl-Jensen[1,7], and Olaf Eisen[2,8]

[*]these authors contributed equally to this work
[1]Physics of Ice, Climate and Earth, Niels Bohr Institute, University of Copenhagen, Copenhagen, Denmark
[2]Alfred-Wegener-Institut Helmholtz-Zentrum für Polar-und Meeresforschung, Bremerhaven, Germany
[3]Remote Sensing Center, University of Alabama, Tuscaloosa, AL, USA
[4]University Grenoble Alpes, CNRS, IRD, IGE, Grenoble, France
[5]British Antarctic Survey, Natural Environment Research Council, Cambridge, UK
[6]Department of Science, University Roma Tre, Rome, Italy
[7]Centre for Earth Observation Science, University of Manitoba, Winnipeg, MB, Canada
[8]Department of Geosciences, University of Bremen, Bremen, Germany

**Correspondence:** David Lilien <david.lilien@nbi.ku.dk>

**Abstract.** The area near Dome C, East Antarctica, is thought to be one of the most promising targets for recovering a continuous ice-core record spanning more than a million years. The European Beyond EPICA consortium has selected Little Dome C, an area ∼35 km south-east of Concordia Station, to attempt to recover such a record. Here, we present the results of the final ice-penetrating radar survey used to refine the exact drill site. These data were acquired during the 2019-2020 Austral summer
using a new, multi-channel high-resolution VHF radar operating in the frequency range of 170-230 MHz. This new instrument is able to detect reflections in the near-basal region, where previous surveys were unable to trace continuous horizons. The radar stratigraphy is used to transfer the timescale of the EPICA Dome C ice core (EDC) to the area of Little Dome C, using radar isochrones dating back past 600 ka. We use these data to derive the expected depth–age relationship through the ice column at the now-chosen drill site, termed BELDC. These new data indicate that the ice at BELDC is considerably older than that at
EDC at the same depth, and that there is about 375 m of ice older than 600 ka at BELDC. Stratigraphy is well preserved to 2565 m, below which there is a basal unit with unknown properties. A simple ice flow model tuned to the isochrones suggests ages likely reach 1.5 Ma near 2525 m, ∼40 m above the basal unit and ∼240 m above the bed, with sufficient resolution ($14\pm1$ ka m$^{-1}$) to resolve 41 ka glacial cycles.

## 1   Introduction

Ice cores provide one of the best records of paleoclimate on 100-ka timescales, but to date no continuous ice core has been recovered that spans more than 800 ka in stratigraphic order. There is great interest in extending ice core records beyond the mid-Pleistocene transition (MPT; 1.25 to 0.7 Ma), since this may provide unique insight in the mechanism which caused the switch between 41- and 100-ka ice-age cycles. An ice core spanning the last ∼1.5 Ma would extend into the period





characterized by regular 41-ka cycles, and would provide a more precise record of greenhouse gases through this transition
than is currently available (Fischer et al., 2013). Several nations or consortia of nations are endeavoring to recover such cores in
East Antarctica as part of the International Partnerships in Ice Core Sciences (IPICS; Beyond EPICA near Dome C, Australia
near Dome C, China near Dome A, Japan near Dome F, Russia near Ridge B, and the US in the Allan Hills and exploring other
potential sites).

The EPICA Dome C ice core (EDC; EPICA Community Members, 2004), drilled at the location now occupied by Concordia
Station in East Antarctica, provides the oldest stratigraphic ice-core climate record recovered to date. The site's cold conditions,
low accumulation, and thickness are conducive to preserving old ice. However, slight melting at the bed suggests that a nearby
site with slightly thinner ice, and thus no basal melt, could preserve a longer record. Ideally, that site would have relatively
smooth bed topography to prevent flow-induced disturbances. Modeling identified two candidate targets in the area (Parrenin
et al., 2017), and subsequent work (Passalacqua et al., 2018; Young et al., 2017) narrowed the search to an area ∼35 km
southwest termed Little Dome C (Figure 1). To obtain the oldest ice at maximum resolution, the core would ideally be at a
location where the ice was as thick as possible without allowing basal melting. While the minimum ice thickness to allow
melting varies spatially with accumulation, ice flow, and geothermal heat flux, several constraints are available for the region.
Analysis of airborne radar data identified a number of subglacial lakes, the shallowest of which lies beneath 2875 m of ice
(Young et al., 2017). Though not framed specifically in terms of minimum ice thickness to cause melting, several thermal
modeling studies of the area (Passalacqua et al., 2018; Van Liefferinge et al., 2018; Parrenin et al., 2017) suggest that parts of
LDC with ice thickness around 2700 m are likely free of basal melt.

Extensive radar work has been conducted in the area of LDC in the frame of Beyond EPICA, which greatly narrowed the area
this present work examined. The initial, aerial survey (Young et al., 2017) mapped the bedrock extensively, greatly improving
the knowledge of the bed compared to the single Operation Ice Bridge flight line in the area. These results also allowed further
inference of basal conditions (Passalacqua et al., 2017) as well as the accumulation rate in the area over the last 73 ka (Cavitte
et al., 2018). Those results led to a targeted, ground-based survey using an impulse radar operating in the 1–5 MHz range (R.
Mulvaney, unpublished), which narrowed the core location to an ∼8 km² area. While modeling shows that LDC is likely to
have old ice, those previous radar surveys were not able to connect any isochrones older than ∼400 ka throughout the area
to the EDC ice core, leaving large uncertainties in the ages nearer the bed. This inability to trace older horizons stems from
a near-basal region that is common in Antarctica, previously described as the "echo-free zone". The cause of the echo-free
zone is unclear, and has been variously attributed to a sharp thermal transition; folding or buckling; and re-circulation and
re-crystallization (see Drews et al. (2009) for a detailed discussion). Indeed, the existence of an echo-free zone is disputed, as
it may simply be an artifact of radar system detection limits. Regardless of the cause, the lack of reflections near the bed in
prior surveys of LDC limited constraints on very old ice in the region.

Here, we present the results of an additional radar survey designed to connect the stratigraphy of that site with the EDC core
and to identify the area at LDC with the highest potential for old ice. This survey utilized a new, highly sensitive radar, which
allowed for detection of older horizons nearer the bed. We focus only on the age constraints provided by this new radar survey
for the chosen ice-core site, Beyond EPICA LDC (BELDC), by comparing the stratigraphy to EDC.

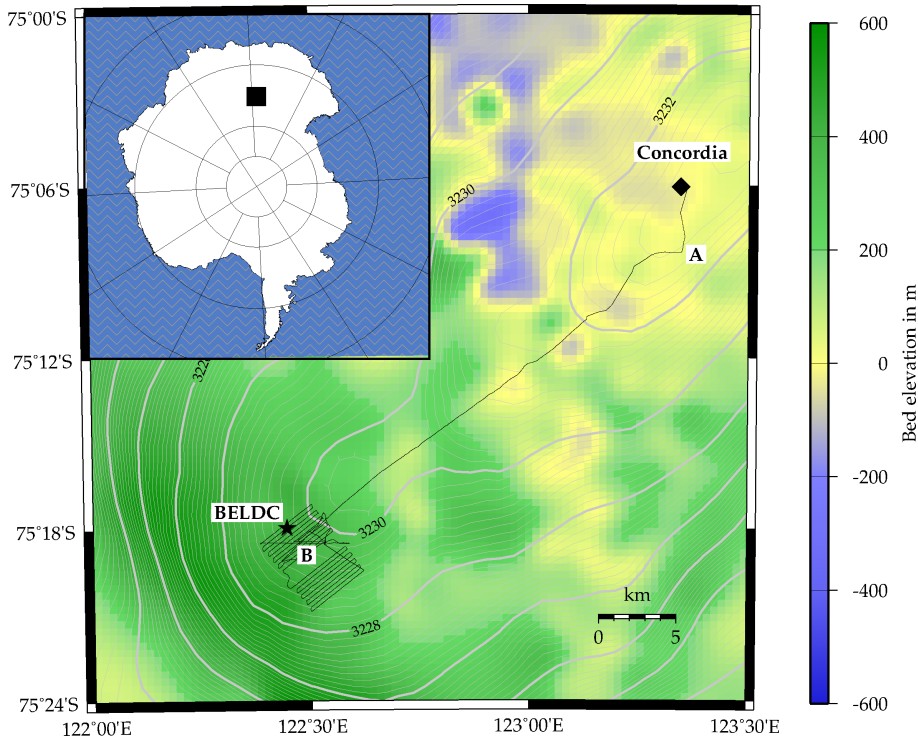

**Figure 1.** Map of study area, with inset showing location in Antarctica. Black lines show radar profiles acquired in 2019. EDC core coincides with Concordia. Contours show surface elevation from Helm et al. (2014). Background colors are bed elevation relative to sea level from Bedmachine v2 (Morlighem, 2020; Morlighem et al., 2020)

## 2   Methods

### 2.1   Data collection and processing

Data were collected using a new very high frequency (VHF) radar, built by the Remote Sensing Center at the University of Alabama (Yan et al., 2020). The system was configured to transmit 8 $\mu$s chirps, with 200 MHz center frequency and 60 MHz bandwidth. Peak transmit power was varied from 125–250 W through the campaign to maximize the signal-to-noise while limiting problems with radio-frequency interference. The system has eight transmit and receive channels, paired with eight monostatic antennas. Due to the logistical challenges of the operating environment, the number of channels in use varied from 5–8. The system was pulled behind a tracked vehicle, with controlling electronics in the rear passenger compartment and antennas approximately 12 m behind. The antennas were set-up such the electric field polarization was oriented across track, above a single sheet of plywood for stiffness and thin PVC mat for slipperiness. Data were collected at travel speeds of 7–12 m s$^{-1}$ over the course of a week in November and December, 2019.





Data processing consisted of coherent integration (i.e. unfocused SAR), pulse compression, motion compensation (by track-
ing internal horizons), coherent channel combination, and de-speckling using a median filter. Two-way travel time was con-
verted to depth assuming a correction of 10 m of firn-air and a constant radar wave speed in ice of 168.5 m $\mu s^{-1}$ (Winter et al.,
2017). After other processing was complete, different radargrams were spliced together to create a continuous profile extending
from EDC to BELDC, and then the data were interpolated to have constant, 10-m horizontal spacing. The re-interpolated data
were used for horizon tracing, which was done semi-automatically to follow amplitude peaks between user-defined clicks. For
the bed reflection, there were often weak, diffuse events shallower than a clear return. We always picked the first notable return
in the region of the bed, so ice-thickness estimates are likely biased shallow; the number of hyperbolic and diffuse events,
likely originating from roughness of the ice-bed interface or physical changes in the lower parts of the ice sheet, would cause
a high risk of misinterpretation with other approaches.

**2.2   Horizon dating and depth–age reconstruction**

Radar reflections were dated by interpolating from the AICC2012 timescale (Veres et al., 2013; Bazin et al., 2013) at the point
of closest approach to the EDC drill site. The radar line ended approximately 100 m horizontally from the EDC borehole. The
depth of the bed reflection there is 3238 m (38.32 $\mu s$), very close to the depths found by Winter et al. (2017) but shallower
than the actual 3260-m depth of the EDC borehole (Parrenin et al., 2007). This offset is likely due to a combination of debris
in the ice causing a too-shallow reflection, small differences in topography over the 100 m offset, and uncertainty in firn-air
content and wave speed. Regardless of the cause, we either must re-scale the thickness to match EDC, or leave it as measured.
We choose the latter since any re-scaling would be highly uncertain.

The dating uncertainty has two primary components: uncertainty in the ice-core timescale and uncertainty in the radar-
horizon depth. The horizon-depth uncertainty can be further subdivided into the component caused by the radargram not
extending exactly to the EDC core site and the component caused by the firn correction and dielectric constant, which affects
each radar trace (see Winter et al. (2017) for a detailed discussion of the components of the error). For the ice-core uncertainty,
we use the previously published estimates from the chronology (AICC2012; Bazin et al., 2013; Veres et al., 2013). We estimated
slope-induced uncertainty from the ∼100 m offset of the radargram from the core using the each horizon's average slope;
slopes ranged from 10 to 60 m km$^{-1}$, resulting in depth uncertainty of 1 to 6 m, increasing with depth. The depth uncertainty
introduced by anisotropy and temperature affecting the dielectric constant is taken to be 1%, and we assume an additional
3-m uncertainty in the firn-air correction. The formal quarter-wavelength uncertainty of the horizon position is small (0.2 m)
compared to other terms. Thus, total depth uncertainties range from 11 m for the upper horizons to 31 m for the lower horizons,
introducing age uncertainties of 1 to 33 ka (found using the depth gradient of the depth–age scale following Winter et al.
(2017)). Combining with uncertainties in the timescale itself, total age uncertainties increase from 2 ka in shallow horizons to
34 ka at depth. The main components of the age uncertainty are all correlated, though we are unable to quantify the extent of
this correlation.





## 3   Results

The processed radargram shows a clear bed reflection and a number of horizons that can be continuously traced from EDC
to LDC (Figure 2). We traced a subset of the visible horizons, selected to span all depths with a concentration in the deepest
areas. In addition to the continuous horizons, there were some that could be identified near both ends of the radargram, where
horizon slopes are relatively flat, but not in the middle. We also traced these partial horizons where possible, to allow more
complete connection of the core's timescale to the BELDC site. The data indicate that the thickness at the chosen site at LDC
is 2764±20 m, with the top 2565 ±20 m showing continuous stratigraphy. Around 2565 m there is a diffuse event that suggests
ice below this depth has different properties; this reflector is discussed further in Section 4.1. The ice thickness is >100 m
shallower than the shallowest subglacial lake that has been observed in the area (Young et al., 2017), and there is no non-bed
parallel down-warping of englacial horizons, both indicating that the site is free of basal melt. To the east of LDC (near km 14
and 27 in Figure 2) there is some down-warping of englacial horizons, but the ice in that area is thicker than at BELDC as the
bed deepens in subglacial valleys.

   At the intersection with EDC, horizons were dated from 71 to 616 ka (Figure 2). Of these, six isochrones were older than
400 ka, the age of the oldest previously traced isochrone in the area, with the oldest being 616 ka. This >50% increase in the
age of dated isochrones introduces a significantly tighter constraint upon the age and age resolution at the LDC site than was
previously available. These data indicate that ice reaches 600 ka at a relatively shallow 2373±20 m at the chosen BELDC site.
The older radar ages lead us to infer ice older than 1.5 Ma, with ∼41 ka m$^{-1}$ as detailed in Section 4.2.

## 4   Discussion

We first discuss a thick basal unit at LDC that may affect the depth–age relationship, before focusing on the depth–age distri-
bution at the BELDC site.

### 4.1   Basal unit

At LDC, there is a notable diffuse event in the radargram around 2500 m depth (pink line in Figure 2). While some events
arrive later than this event, and thus most likely originate deeper, there are no continuous or coherent reflecting horizons below
this depth. However, this reflection is at a depth where the radar is capable of imaging continuous reflections throughout most
of the radar profiles, suggesting a change in ice properties rather than a system detection limit. The origin of this reflection
may be the result of a sharp transition in crystal fabric, heterogeneous small-scale roughness, stagnant ice, tightly spaced
or disrupeted isochrones, or perhaps some other relic feature; regardless of its origin, the lack of stratigraphy suggests that
recovery of a climate record below this depth may be difficult, potentially impossible. Thus, in subsequent analysis we follow
a conservative approach and consider the maximum recoverable age both assuming that this horizon marks the deepest useful
climate information and assuming that useful information continues below this horizon. We emphasize that despite of the lack

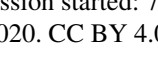



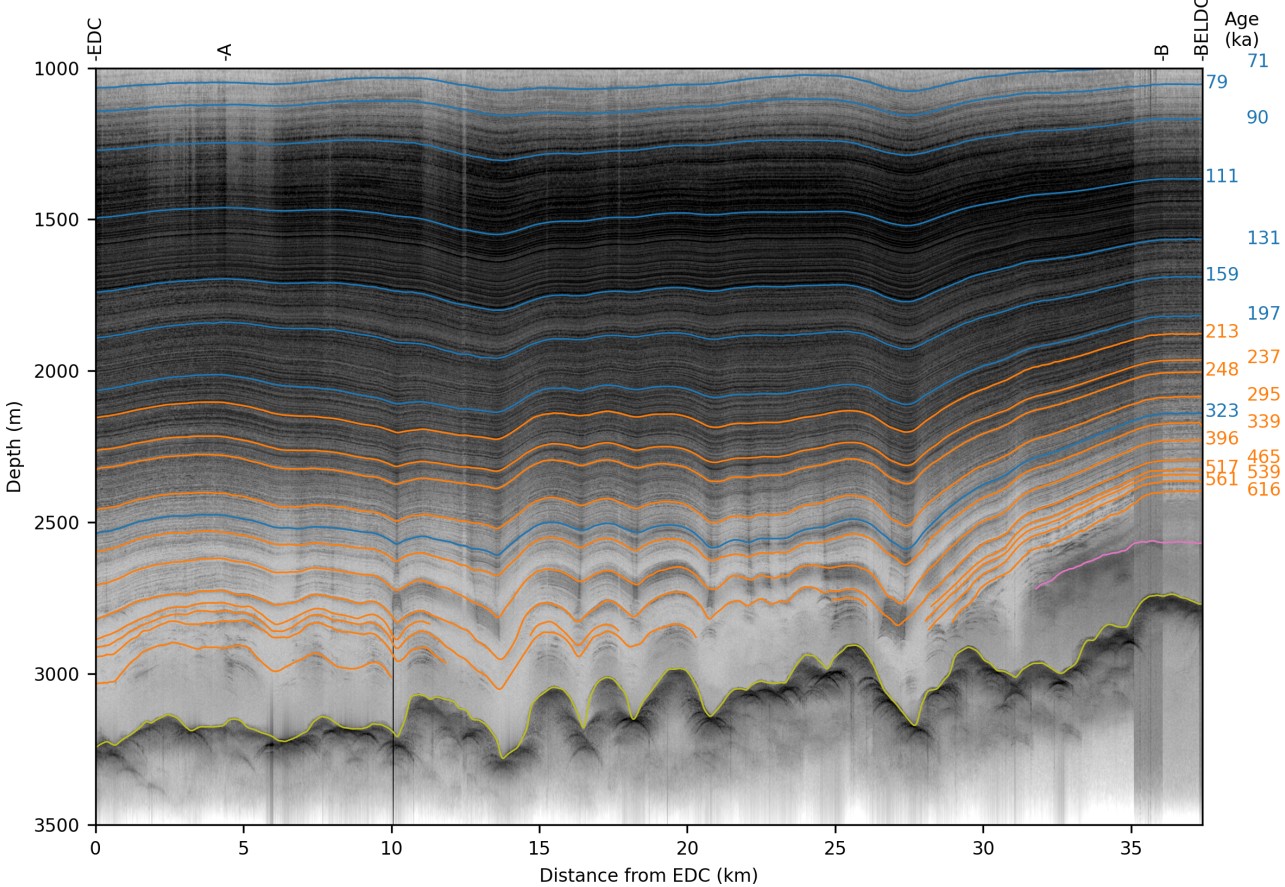

**Figure 2.** Radargram extending from EDC to the chosen ice-core site BELDC at Little Dome C. Horizons in blue were identified by Winter et al. (2017), using other radar systems, with additional horizons traced in orange. Pink marks the top of the basal unit at LDC (between km 32 and 38) and yellow denotes the bed reflection. Horizon ages are shown at right.

of continuous radar isochrones, studies at other ice-core sites (e.g. EDC and EDML) showed that climate information might still be retrievable at least in the top part of basal units with similar characteristics (Ruth et al., 2007; Tison et al., 2015, e.g.,).

## 4.2 Depth–age at the BELDC site

While previous work has used sophisticated models to make estimates of the depth–age scale at LDC (Parrenin et al., 2017), here we seek a more simple constraint relying almost solely on the radar data. We fit a Dansgaard-Johnsen model (Dansgaard and Johnsen, 1969) to the horizons at the chosen core site. The model takes the form

$$
\bar{t} =
\begin{cases}
\frac{2H-h}{2a} \ln\left(\frac{2(H-d)-h}{2H-h}\right) & \text{if } d < (H-h) \\
\frac{2H-h}{a}\left(\frac{h}{H-d}-1\right) + \frac{2H-h}{2a}\ln\left(\frac{h}{2H-h}\right) & \text{if } d \geq (H-h)
\end{cases}
\tag{1}
$$





where $\bar{t}$ is the age in steady state, $d$ depth, $a$ accumulation, $H$ ice thickness, and $h$ the kink height below which horizontal
deformation is concentrated. This model is simple, but has a long history of successful application to ice-core glaciology (e.g.,
Dansgaard and Johnsen, 1969; Dahl-Jensen et al., 1999; Winski et al., 2019). As in Parrenin et al. (2017), we used a temporally
variable accumulation rate, and solved for the depth-age using a pseudo-steady method which permits analytical solutions even
with the temporally variable accumulation (Parrenin et al., 2006). This involves a simple change of variable between time, $t$,
and steady time, $\bar{t}$ of the form

$$\bar{t} = \int_0^t R(t')dt', \tag{2}$$

where $t'$ is a dummy variable for integration and $R(t) = a_E(t)/\bar{a}_E$ is the normalized accumulation at a given time of the EDC
record (Bazin et al., 2013). Equation 2 defines a bijection between $t$ and $\bar{t}$, so we can first find the steady-state Dansgaard-
Johnsen profile using Equation 1, and then convert to the equivalent profile incorporating the EDC accumulation variations
using Equation 2. In this formulation, the temporally variable accumulation enters only as the non-dimensional scaling, $R(t)$,
while $a$ in Equation 1 is treated as a constant.

We used a Markov-chain Monte Carlo method, implemented with PyMC3 (Salvatier et al., 2016), to find the probability
distribution of the resulting depth–age scale by varying $a$, $H$, and $h$. The depth–age scale from a random sample of parameters
drawn from the posterior distributions is shown the gray histogram in Figure 3b. The best fit ice thickness is 2650 m $\pm$22 m,
which falls in the midst of the basal unit (uncertainties in this section are the standard deviation of the distribution of modeled
depth–age profiles). The mean accumulation is $17.0\pm0.2$ kg m$^{-2}$ a$^{-1}$, equal within error to the 16.9 kg m$^{-2}$ a$^{-1}$ average
at EDC (Bazin et al., 2013). The mean kink height is $735\pm71$m. The relatively small effective ice thickness suggests that the
basal unit is partially stagnant, or flows much more slowly than the overlying ice, such that the deformation of the overlying ice
column is unaffected by this deeper ice and ages asymptote within the basal unit. The corresponding depth–age scales reach
1.5 Ma at $2524\pm12$ m, with average age resolution of $14\pm1$ ka m$^{-1}$, suggesting that resolution is sufficient for measuring
41-ka glacial cycles at this age. The original BEOI resolution target, 20 ka m$^{-1}$ (Fischer et al., 2013), is passed at $2545\pm17$ m
and age $1850\pm76$ ka, within error of the top of the basal unit. Further refinement of the depth–age scale with more sophisticated
models and elaborate assumptions could refine the estimated age and age resolution, but will likely result in similar values.

## 5   Conclusions

Newly collected radar data provide a tighter constraint on the depth–age scale at LDC. These data reveal traceable stratigraphy
in ice >600 ka old in the region (~50% older than previously available). Near LDC, they also indicate a unit of basal ice in
which few events are visible; the origin of this basal unit requires further investigations as its flow properties and composition
are unknown. The stratigraphy indicates that old ice lies much shallower at BELDC than at EDC. A Dansgaard-Johnsen model
of the depth–age scale, fitted to the isochrone data, indicates that 1.5 Ma-old ice lies at ~2525 m depth, where stratigraphy is
still in tact, and preserved with ~14 ka m$^{-1}$ resolution. Very old (>1.5 Ma) ice could exist atop the basal unit, which appears

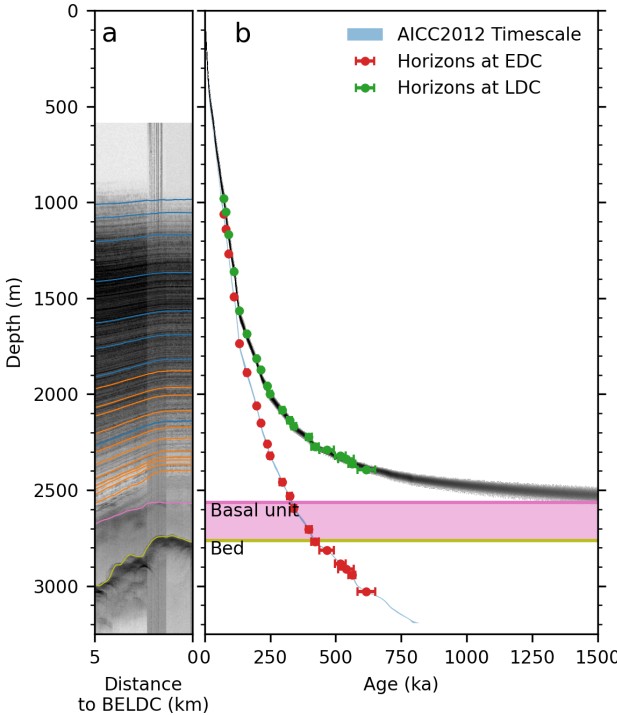

**Figure 3. a** Radargram near BELDC. **b** Depth–age scale at BELDC and EDC (as in Figure 2). Blue line shows AICC2012 chronology (Bazin et al., 2013; Veres et al., 2013). Red dots show radar horizons in their nearest 100 m to EDC, while green dots show the same for the 100 m nearest BELDC; error bars indicate estimated age uncertainty as described in the text. Gray shaded regions shows the histogram of the Monte-Carlo simulations using a Dansgaard-Johnsen model; darker colors indicate greater likelihood. Pink shaded region marks the basal unit and yellow line indicates the bed, both at BELDC.

partially stagnant and presumably also contains >1.5 Ma ice, though the lack of stratigraphy does not allow firm conclusions to which extent useful climatic information may or may not be preserved below 2565 m depth.

*Data availability.* The radar profile displayed in Figure 2 will be made available on pangaea.de after publication.

*Author contributions.* All authors contributed to survey design. PG, JY, and CO designed and built the radar system. PG and JY led the development of the processor. DS, DT, and DL collected the radar data. DL processed and traced the radar data and implemented the depth–
age model. FP, CR, DDJ, OE, and DS also contributed to the age modeling. DL and DS wrote the first draft of the manuscript. All authors contributed to writing the final manuscript.



*Competing interests.* OE is CEIC and CM is an E of TC. The authors declare no other competing interests.

*Acknowledgements.* This publication was generated in the frame of Beyond EPICA. The project has received funding from the European Union's Horizon 2020 research and innovation programme under grant agreement No. 815384 (Oldest Ice Core). It is supported by national

partners and funding agencies in Belgium, Denmark, France, Germany, Italy, Norway, Sweden, Switzerland, The Netherlands and the United Kingdom. Logistic support is mainly provided by PNRA and IPEV through the Concordia Station system. The radar shipment and personnel transportation to Antarctica was provided by U.S. NSF under grant 1921418. The development of the VHF radar was supported by the University of Alabama. The opinions expressed and arguments employed herein do not necessarily reflect the official views of the European Union funding agency or other national funding bodies. This is Beyond EPICA publication number XX. We thank Saverio Panichi and

Michele Scalet, as well as the Concordia station team, for support in the field. We also thank the U.S. National Guard and Royal New Zealand Air Force for the flights between Christchurch and McMurdo. We are grateful also for the logistical support from McMurdo and Zucchelli stations.



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
