# Peer review of "Brief Communication: New radar constraints support presence of ice older than 1.5 Ma at Little Dome C"

_The Cryosphere, 2020_

## Referee Comment (RC1) · Lucas Beem (Referee) · 21 Dec 2020

The authors report a new ground based ice penetrating radar study of a potential ice core site. The new observations resolve internal reflecting horizons at a greater depth than previously observed. Tracing these horizons from the dated EDC core extends dated horizons as old as ∼600 ka into the study region. These observations give important new constraints on depth age modeling for ice older than the dated horizons. The authors use a modified Dansgaard-Johnson model to determine that stratigraphically intact ice dated to older than 1.5 Ma may exist at this site.

This is a fine contribution and is timely given the ongoing efforts to recover a mid-

Pleistocene transition spanning ice core. I have one point of concern and some additional small comments.

On line 100 and 101 the authors state 'In addition to the continuous horizons, there were some that could be identified near both ends of the radargram, where horizon slopes are relatively flat, but not in the middle. We also traced these partial horizons where possible.' How is it known that the same horizon is traced on both sides of a visibility gap? Under what conditions is it 'possible'? Figure 2 appears to show some examples of tracing over a visibility gap. These gaps included significant variation in horizon slope, is this what is meant by 'relatively flat'? This methodology should be described in more detail as well as its impact on horizon age uncertainty. Perhaps the horizons which were traced over gaps in visibility, which I presume tend to be the deepest horizons, are not suitable for tracing. I cannot evaluate this potential as I do not understand what is done. Which horizons are not completely continuous from EDC to BELDC? The reader should be given more information about the practice used and its impact on analysis/conclusions. For instance, what happens if, say, the 6 deepest horizons are excluded from the analysis or all horizons that have gaps.

Model uncertainty seems to be understated. Line 156 appears to be the only location where specific model age error is stated (+/- 76 ka), which is about 5% of the magnitude. This seems to be significantly less than what is depicted in Figure 3b. If I trace a horizontal line at depth 2500 (the approximate depth in which the authors state 1.5 Ma is reached) the gray shaded region extends from $\sim$1 Ma to some value greater than 1.5 Ma. Seems to me that the model results suggest that at a depth of 2500m the age would be some value greater than 1 Ma, with a range that extends to beyond 1.5 Ma. Its full range is not depicted. The figure seems inconsistent with a stated deviation of +/- 76 ka.

Because 1.5 Ma is an age discussed in the manuscript, perhaps figure 3b x-axis should extend beyond this magnitude.

How the specific BELDC location was identified as the most suitable for the location of a bore hole (as opposed to others within the new survey) could be described in more detail, or a citation added. The reader would like to know it is relevant to apply the model in this location.

L33: Shallowest lake? I don't believe lake depths are known. Perhaps the authors mean beneath the least amount of ice?

L78: report the value from Winter et al. and where those depths located (at the core, at the end of the new line, both?, elsewhere?)

L105: Referring to lakes as shallow when depth of ice overburden is being discussed unnecessarily confuses terms.

L126: extra 'of'. Despite of the lack

L128: 'e.g.,)' in citation

L129: section 4.2, Is this section mostly methods?

---

## Referee Comment (RC2) · Xiangbin Cui (Referee) · 16 Jan 2021

General Comments

With very deep layers captured by the grounded ice penetrating radar survey at the Little Dome C, this work further confirms the existence of an old ice there extends into the past of 1.5 Ma through a modified D-J model with the new constraints of the radar data. It shows great values on evaluating deep ice age and locating a deep ice core for the oldest ice. \

Specific Comments

[Figure]

Lines 21 – 23, can the authors give some more references here? It should help people to extend and understand recent progresses on searching for the oldest ice in Antarctica by the international groups (e.g. for Dome A, Liyun Zhao, et al., 2018, Where is the 1-million-year-old ice at Dome A; for Dome F, Nanna B. Karlsson, et al., 2018, Glaciological characteristics in the Dome Fuji region and new assessment for "Oldest Ice"; for Ridge B, Xiangbin Cui et al., 2020, The Scientific Operations of Snow Eagle 601 in Antarctica in the Past Five Austral Seasons; for Titan Dome, Lucas H Beem, et al., 2020, Characterization of Titan Dome, East Antarctica, and potential as an ice core target; or others more suitable)

Does the star in Fig.1 note the position of BELDC? It should be better to highlight the profile of AB (Fig. 2) in Fig.1 with thick black lines or color lines. The radar profile presented in Fig. 2 should not a direct line between EDC and BELDC, and should have many turns. In addition, too many lines are placed in a very small area, so the authors may need zoom in the area to show clearly the distribution of the radar lines.

If there's no figure number limitation or page limitation, please provide a field operation figure.

Line 104-106, the bed reflection coefficient should be analyzed to verify the basal melt. "... the shallowest subglacial lake..."should mean the buried depth of the lake is the smallest.

Figure2, I am not clear how to know the age of the deep layers with 517, 539, 561, and 616 ka, because there's no link between these layers with EDC.

Technical corrections

Line 164, should be "intact"

---

## Short Comment (SC1) · 1 Feb 2021

Short Comment of Brief Communication: New radar constraints support presence of ice older than 1.5 Ma at Little Dome C by Lilien et al.

I would like to bring up a few additional points to the RCs submitted.

-I find that the description of radar and modeling work done prior to this study is missing several seminal studies:

1. the Van Liefferinge and Pattyn (2013) study is never mentioned, in particular in the introduction. However, it was their modeling work that identified promising oldest ice

sites all over the East Antarctic Ice Sheet that then determined the sites (and so LDC) that should be further surveyed. This should be discussed in the paper, in particular in the introduction.

2. In the discussion of radar depth/age uncertainties, only the Winter et al (2017) study is cited. However, Cavitte et al. (2016) laid a lot of the ground work (and cited by Winter al (2017)). In addition, there is also the work of MacGregor et al (2015) that should cited here.

-Line 39-40: Van Liefferinge et al. (2018) also used the airborne radar data to refine their basal conditions.

-Line 48-49 (introduction) and Section 4.1: a diffuse reflector at depth in the radar data was already observed in the previous radar surveys, both in the airborne OIA data and the GPR DELORES survey data. This was mostly discussed during BE-OI meetings (which could perhaps be mentioned as BE-OI pers. comm.). Some of it was included in the publicly available thesis Cavitte et al (2017) available here: https://repositories.lib.utexas.edu/handle/2152/62593

-Section 4.2: why use such a simple model for dating when there's a 1D inverse model that has shown it works really well in the region (Parrenin et al., 2017 as cited), and Fred Parrenin is one of the co-authors of this paper?

-I wonder how the deepest isochrones traced in the manuscript were dated as they seem to get cut off around 30 km from the EDC site.

-This was not out at the time of submission but there is now a paper (Cavitte et al., 2021) in ESSD Discussions that summarizes all the isochrones that have been traced in the region using the UTIG/OIA/DELORES data sets. It shows that a number of isochrones that are older than 400 ka could be traced in these radar surveys. These deepest isochrones could not be traced to the EDC site due to the steep topography of the Concordia Subglacial Trench, but were dated using the Parrenin et al. (2017) inverse model. The deepest two isochrones in Cavitte et al (2021) show ages of ∼610+/-35 ka and ∼709+/-54 ka. So, similar ages to those obtained at depth in the manuscript under review. It would be great to see this work referenced and included in the discussion of this manuscript.

Best,

Marie Cavitte

References mentioned: - Van Liefferinge, B. and Pattyn, F.: Using ice-flow models to evaluate potential sites of million year-old ice in Antarctica, Clim. Past, 9, 2335–2345, https://doi.org/10.5194/cp-9-2335-2013, 2013. -Cavitte, M.G., Blankenship, D.D., Young, D.A., Schroeder, D.M., Parrenin, F., Lemeur, E., Macgregor, J.A. and Siegert, M.J., 2016. Deep radiostratigraphy of the East Antarctic plateau: connecting the Dome C and Vostok ice core sites. Journal of Glaciology, 62(232), pp.323-334. -MacGregor JA and 9 others (2015) Radiostratigraphy and age structure of the Greenland Ice Sheet. J. Geophys. Res.: Earth Surf., 120 (2), 212–241. -Cavitte, M. G. P., Young, D. A., Mulvaney, R., Ritz, C., Greenbaum, J. S., Ng, G., Kempf, S. D., Quartini, E., Muldoon, G. R., Paden, J., Frezzotti, M., Roberts, J. L., Tozer, C. R., Schroeder, D. M., and Blankenship, D. D.: A detailed radiostratigraphic data set for the central East Antarctic Plateau spanning the last half million years, Earth Syst. Sci. Data Discuss. [preprint], https://doi.org/10.5194/essd-2020-393, in review, 2020.

---

## Editor Comment (EC1) · Joseph MacGregor (Editor) · 2 Feb 2021

Hi Dr. Lilien et al.,

I've now received effectively three reviews of your Brief Communication submission. These are mostly positive and highlight the value of your contribution and the context within the broader Oldest Ice effort and other studies in the region.

In general, I agree with their comments, and find this submission responsive to the spirit of TC Brief Communication. I am in particular concurrence with reviewer #1's comment on 4.2 being more methods than discussion (so consider moving part of it),

reviewer #1 and #2's concern over the terms used to describe subglacial lakes, and #3's concern over the nature of the depth-age model used (not sure it needs to be changed necessarily, but context suggested by reviewer #3 is relevant). As is probably not surprising given my Greenland studies, I am somewhat more comfortable with the matching of discontinuous reflections than a couple of the reviewers. However, given that this study is effectively centered around a single radargram, a couple of zoom-ins on the deep ends of near each core, appended to Figure 2 could help make the case based on reflectivity patterns that they are indeed the same.

Several referees suggested additional citations, many of which seem appropriate, but this leads to a broader challenge with the nature of the Brief Discussion format, which is intended to limit length and has several specific metrics, including number of references:

https://www.the-cryosphere.net/about/manuscript_types.html

It is apparently my responsibility to enforce compliance with these metrics. I ask that you not exceed any metric for a TC Brief Communication by more than ~25%, and to not add any figures. In particular, this will require you to be parsimonious with references (24 in original submission). Given many of the suggestions seem sensible, this may require reducing existing ones. "(e.g., X)" is often fine instead "(X, Y, Z)".

Below are a few comments I noted during my re-reading of the MS:

- Define LDC acronym. - 39: IceBridge - 42: R. Mulvaney is a co-author, so this citation is a bit odd. Is there a conference abstract that could be referenced instead? - 93: "depth gradient of age following Winter et al." seems simpler and still correct - 95: "34 ka at depth"...what "depth" in this context? - 95: Precisely which components of the age uncertainty are meant here? Also, are they vertically or horizontally correlated? The latter seems a safe assumption, but perhaps the former is what the authors are intending to highlight? - 103: Here it would be helpful to highlight the fraction of ice thickness that 2565/2764 represents, because frankly it is an impressive number

compared to most radargrams. 103: "event" is evocative but confusing in this context (fast-time? a particularly paleoclimatic transition?). Better would be "reflector" IMO. - Figure 2: I highly recommend converting tedious captions into legends whenever possible, and that seems worthwhile here. - Copernicus formatting uses the abbreviation "Sect." for "Section". https://www.the-cryosphere.net/submission.html

Regards,

Joe MacGregor NASA/GSFC

---

## Author Response (AR1)

**Response to reviews of Brief Communication: New radar constraints support presence of ice older than 1.5 Ma at Little Dome C**

We thank Drs. Beem and Cui for their invited and very constructive reviews, Dr. Cavitte for her constructive comments on the manuscript, and Dr. MacGregor for providing clear and prompt editorial guidance. The editor's comments, and the reviews are, included below, in black. Our responses are in blue, and the corresponding alterations we have made to the text in *blue italics*. All three reviewers commented on the use of discontinuous radar horizons, and we address that issue collectively before addressing the individual reviews:

The four deepest horizon (dated to 517, 539, 561, and 616 ka at EDC) are not continuous across the echogram, but were used in reconstructing the age at LDC. The deep stratigraphy is of sufficient quality that we feel that we can connect the pattern of the horizons on either side of gaps. We would argue that almost all radar practitioners trace over gaps, though the exact length of gap that different authors will trace varies. Perhaps this survey jumps larger gaps than most, and we have tried to better incorporate the resulting uncertainty, and to present clearer information for the reader to judge for themselves the quality of the deep stratigraphy.

We have taken several steps to make this more clear:
1. We have added zoomed-in, contrast-adjust panels to Figure 2 to show these horizons in flat areas near each end, so the readers can see the stratigraphy being used (shown below).
2. We have removed the deepest horizon, which was dated to 616 ka, since the reflection pattern is less distinctive. Upon re-examination, there were two possible ways to trace this horizon near EDC, so we felt it was best to remove it. The other three discontinuous horizons do not have this issue.
3. We have added a second set of error bars to Figure 3, showing the spread of model results when the discontinuous horizons (i.e. the three deepest horizons) are excluded (shown below). The uncertainty is of course wider, and it demonstrates the results with the additional horizons included merely add confidence that the ages lie in a subset of those results. Where ages are referenced in the text, we have added parentheticals giving the values excluding these three horizons. We note that this also required switching the visualization in Figure 3b to use the 95% confidence and 1-$\sigma$ bounds rather than a histogram so that everything is legible.
4. We have altered the text to clarify this thought process (changes throughout).

The effect on the results is notable but does not change our conclusions. The 95% confidence interval of the depth-age scales including constraints from the discontinuous horizons is a subset of the 95% confidence interval without those horizons. In other words, they narrow the uncertainty but do not fundamentally alter the results. However, the discontinuous horizons do pull the distribution toward being slightly younger, and thus towards having slightly better depth-age resolution. This is now discussed in Sec. 4.2. We have left the values in the abstract and conclusion as those including the discontinuous horizons.

[Figure]

**Response to editor comment by Joe MacGregor**

Hi Dr. Lilien et al., I've now received effectively three reviews of your Brief Communication submission. These are mostly positive and highlight the value of your contribution and the context within the broader Oldest Ice effort and other studies in the region.

Thank you for the positive comments, and for your continued consideration of this manuscript. We have briefly pointed out how we have addressed your comments, with full details found in our response to the other reviews.

In general, I agree with their comments, and find this submission responsive to the spirit of TC Brief Communication. I am in particular concurrence with reviewer #1's comment on 4.2 being more methods than discussion (so consider moving part of it),

Indeed, this was misplaced. It is now mostly Sec. 2.3.

reviewer #1 and #2's concern over the terms used to describe subglacial lakes,

We have removed the word shallow since it gave the wrong connotation.

and #3's concern over the nature of the depth-age model used (not sure it needs to be changed necessarily, but context suggested by reviewer #3 is relevant).

We switched to Lliboutry velocity profile as in Parrenin et al., 2017; the other differences compared to that study do not affect the results. Hopefully switching the model obviates the need for lengthy justification/argument. It turned out to be easier just to switch the model than to try to provide a brief yet compelling justification. The depth—age scale was altered, though our conclusions are unchanged.

As is probably not surprising given my Greenland studies, I am somewhat more comfortable with the matching of discontinuous reflections than a couple of the reviewers. However, given that this study is effectively centered around a single radargram, a couple of zoom-ins on the deep ends of near each core, appended to Figure 2 could help make the case based on reflectivity patterns that they are indeed the same.

The full set of changes to support the use of these horizons is described at the top of this document. While we stand by the decision to trace over gaps, and believe the pattern of these reflectors is distinctive, the gaps are perhaps larger than what people usually trace over in Antarctica. We have removed the deepest horizon, which is less clear than the other three discontinuous ones, since on further examination there were two ways it could be traced at EDC. We hope that the combination of zoom-ins, model results with those horizons excluded, and further discussion in the text, can both show readers why this tracing is justifiable and also show the main conclusions are relatively robust to our decision to include those horizons.

Several referees suggested additional citations, many of which seem appropriate, but this leads to a broader challenge with the nature of the Brief Discussion format, which is intended to limit length and has several specific metrics, including number of references:
https://www.the-cryosphere.net/about/manuscript_types.html
It is apparently my responsibility to enforce compliance with these metrics. I ask that you not exceed any metric for a TC Brief Communication by more than ~25%, and to not add any figures. In particular, this will require you to be parsimonious with references (24 in original submission). Given many of the suggestions seem sensible, this may require reducing existing ones. "(e.g., X)" is often fine instead "(X, Y, Z)".

By cutting the least relevant citations in the original manuscript, we have 22 references including much of what the reviewers have requested. The requested references that we did not accommodate were: Cavitte et al., 2016 and MacGregor et al., 2015 referring to radar depth-age uncertainties (we feel that our methodology is adequately justified by Winter et al., 2017) and the references for the other oldest-ice programs (which would have added at least 4 additional references used only in a single sentence).

Below are a few comments I noted during my re-reading of the MS:

- Define LDC acronym.

Done

- 39: IceBridge

Fixed

- 42: R. Mulvaney is a co-author, so this citation is a bit odd. Is there a conference abstract that could be referenced instead?

This was included in the new Cavitte preprint (ESSD), so we reference that.

- 93: "depth gradient of age following Winter et al." seems simpler and still correct

True, changed to this.

- 95: "34 ka at depth"...what "depth" in this context?

Clarified that this is for the deepest horizons.

- 95: Precisely which components of the age uncertainty are meant here? Also, are they vertically or horizontally correlated? The latter seems a safe assumption, but perhaps the former is what the authors are intending to highlight?

We have made this much more explicit to indicate that we were indeed referring to vertical correlation: *"The uncertainties of the horizon ages are correlated with each other, since an incorrect firn-air correction or dielectric constant, or an incorrect age scale at EDC, affects the inferred age of all these horizons, though we are unable to quantify the extent of this correlation."*

- 103: Here it would be helpful to highlight the fraction of ice thickness that 2565/2764 represents, because frankly it is an impressive number compared to most radargrams.

Thanks, we have added that it 2565 m corresponds to ~93% of the thickness. We are of course happy to highlight this!

103: "event" is evocative but confusing in this context (fast-time? a particularly paleoclimatic transition?). Better would be "reflector" IMO.

We see how this is confusing, but it is not clear that this is a reflector. To many geophysicists, "reflector" denotes the (continuous) physical feature at depth, whereas the reflection is a (continuous) signal in the data. An event is usually considered a signal seen in the data, but the origin can vary, i.e. it could be an artifact (e.g. a multiple), a diffraction or indeed reflection (here we follow definition 1 of the SEG dictionary for event). In this particular case, it could be a response to a change in fabric or other physical properties affecting the return power through scattering or interference, and not a reflector (e.g. conductivity horizon) in the traditional sense. We have made this explicit: *Around 2565 m there is a change in the amplitude of the radar returns that suggests ice below this depth has different properties; this feature is discussed further in Sec 4.1.*
And in section 4.1:
*At LDC, there is a notable event (i.e. change in the return power) at around 2565 m depth.*

- Figure 2: I highly recommend converting tedious captions into legends whenever possible, and that seems worthwhile here.

This caption has been moved as much as possible into labels on the figure.

- Copernicus formatting uses the abbreviation "Sect." for "Section". https://www.the-cryosphere.net/submission.html

Fixed.

Regards, Joe MacGregor NASA/GSFC

**Response to review by Lucas Beem**

The authors report a new ground based ice penetrating radar study of a potential ice core site. The new observations resolve internal reflecting horizons at a greater depth than previously observed. Tracing these horizons from the dated EDC core extends dated horizons as old as ~600 ka into the study region. These observations give important new constraints on depth age modeling for ice older than the dated horizons. The authors use a modified Dansgaard-Johnson model to determine that stratigraphically intact ice dated to older than 1.5 Ma may exist at this site.
This is a fine contribution and is timely given the ongoing efforts to recover a mid Pleistocene transition spanning ice core. I have one point of concern and some additional small comments.

On line 100 and 101 the authors state 'In addition to the continuous horizons, there were some that could be identified near both ends of the radargram, where horizon slopes are relatively flat, but not in the middle. We also traced these partial horizons where possible.' How is it known that the same horizon is traced on both sides of a visibility gap? Under what conditions is it 'possible'? Figure 2 appears to show some examples of tracing over a visibility gap. These gaps included significant variation in horizon slope, is this what is meant by 'relatively flat'? This methodology should be described in more detail as well as its impact on horizon age uncertainty. Perhaps the horizons which were traced over gaps in visibility, which I presume tend to be the deepest horizons, are not suitable for tracing. I cannot evaluate this potential as I do not understand what is done. Which horizons are not completely continuous from EDC to BELDC? The reader should be given more information about the practice used and its impact on analysis/conclusions. For instance, what happens if, say, the 6 deepest horizons are excluded from the analysis or all horizons that have gaps.

We have responded to this in detail above in our general reply. It is correct that it is the deepest horizons that are discontinuous--there were four, of which we have removed one. We now show more detailed imagery of those horizons to lead the reader through the tracing. We consider these horizons suitable for tracing, but have provided secondary analysis showing that excluding these horizons does not drastically alter the results.

Model uncertainty seems to be understated. Line 156 appears to be the only location where specific model age error is stated (+/- 76 ka), which is about 5% of the magnitude. This seems to be significantly less than what is depicted in Figure 3b. If I trace a horizontal line at depth 2500 (the approximate depth in which the authors state 1.5 Ma is reached) the gray shaded region extends from ~1 Ma to some value greater than 1.5 Ma. Seems to me that the model results suggest that at a depth of 2500m the age would be some value greater than 1 Ma, with a range that extends to beyond 1.5 Ma. Its full range is not depicted. The figure seems inconsistent with a stated deviation of +/- 76 ka.

This is a good point; the uncertainty becomes a little confusing because the slope of the depth-age curve is so low, and the text implied something we did not intend. The uncertainty in the age referred to when the resolution threshold was crossed, not to the age at that particular depth. The depth-age gradient of the model is well constrained at a given age, though the age at a given depth is not so well constrained. As the reviewer notes, putting both these values in a single sentence implied a tight bound the age at a depth, which is incorrect. We simply deleted the reference to the depth here, since age at the target resolution is the quantity of paleoclimatic

interest. We now avoid giving an age spread at a given depth, since this quantity changes rapidly due to the curvature of the depth-age scale.

Because 1.5 Ma is an age discussed in the manuscript, perhaps figure 3b x-axis should extend beyond this magnitude.
We have extended the x-axis to 1.75 Ma so that 1.5 Ma is fully visible.

How the specific BELDC location was identified as the most suitable for the location of a bore hole (as opposed to others within the new survey) could be described in more detail, or a citation added. The reader would like to know it is relevant to apply the model in this location.
To justify this site, we have added: *The exact location for the ice-core site, Beyond EPICA LDC (BELDC; 75◦17'57.02"S, 122◦26'42.5"E, 3230 m above the WGS84 ellipsoid as of 2020), has deep, flat, visible stratigraphy and lies within the region of LDC identified by previous studies as free of basal melt and likely to contain old ice.*
L33: Shallowest lake? I don't believe lake depths are known. Perhaps the authors mean beneath the least amount of ice?
Indeed this is what we meant. It is now: *all of which lie beneath at least 2875 m of ice*
L78: report the value from Winter et al. and where those depths located (at the core, at the end of the new line, both?, elsewhere?)
Changed to: *within the depths at closest approach to EDC found by other radar systems (approximately 3220–3286 in Winter et al. (2017))*
L105: Referring to lakes as shallow when depth of ice overburden is being discussed unnecessarily confuses terms.
Fixed here as well. *The ice thickness is >100 m less than the minimum thickness over any subglacial lake observed in the area.*
L126: extra 'of'. Despite of the lack
Fixed.
L128: 'e.g.,)' in citation
Fixed.
L129: section 4.2, Is this section mostly methods?
We moved most of this section to the methods. It is now 2.3.

**Response to review by Xiangbin Cui**

General Comments With very deep layers captured by the grounded ice penetrating radar survey at the Little Dome C, this work further confirms the existence of an old ice there extends into the past of 1.5 Ma through a modified D-J model with the new constraints of the radar data. It shows great values on evaluating deep ice age and locating a deep ice core for the oldest ice.

Specific Comments
Lines 21 – 23, can the authors give some more references here? It should help people
to extend and understand recent progresses on searching for the oldest ice in Antarctica by the
international groups (e.g. for Dome A, Liyun Zhao, et al., 2018, Where
is the 1-million-year-old ice at Dome A; for Dome F, Nanna B. Karlsson, et al., 2018,
Glaciological characteristics in the Dome Fuji region and new assessment for "Oldest
Ice"; for Ridge B, Xiangbin Cui et al., 2020, The Scientific Operations of Snow Eagle
601 in Antarctica in the Past Five Austral Seasons; for Titan Dome, Lucas H Beem, et
al., 2020, Characterization of Titan Dome, East Antarctica, and potential as an ice core
target; or others more suitable)
We have not added these due to the limit on references for the format of this submission. Unfortunately, citing a paper for all of these efforts would have taken 20% of our total references for this single sentence, which does not allow us enough room elsewhere. Instead, we have added a short comment to refer the reader to the other papers in the "Oldest Ice" special issue.
*see other articles in this special issue for details on these efforts*

Does the star in Fig.1 note the position of BELDC? It should be better to highlight the
profile of AB (Fig. 2) in Fig.1 with thick black lines or color lines. The radar profile
presented in Fig. 2 should not a direct line between EDC and BELDC, and should
have many turns. In addition, too many lines are placed in a very small area, so the
authors may need zoom in the area to show clearly the distribution of the radar lines.
We have enlarged this figure to make things more clear (see below). The line used for Figure 2 is now in bold. There are indeed three turns on the radar line in Figure 2, which are all marked at the top so that the reader can place this in context--that is now noted in the caption of the figure. The splicing together to create the full line is also noted in the text:
*After other processing was complete, different radargrams were spliced together to create a continuous profile extending from EDC to BELDC*

[Figure]

If there's no figure number limitation or page limitation, please provide a field operation Figure.

Per the editor's instructions, we have not added an additional figure.

Line 104-106, the bed reflection coefficient should be analyzed to verify the basal melt. ". . . the shallowest subglacial lake..."should mean the buried depth of the lake is the Smallest.

We have fixed our language so as not to use "shallowest".

The reflection coefficient analysis has previously been performed on BAS' DELORES data acquired in the same project, for which a publication is being prepared, together with a comprehensive interpretation in the ice-dynamic setting. Results are so far already available in reports (but not peer reviewed). Although we agree that an amplitude analysis is desirable in general, we consider it to go beyond the scope of this manuscript.

Figure2, I am not clear how to know the age of the deep layers with 517, 539, 561, and 616 ka, because there's no link between these layers with EDC.

This is addressed above in our general reply since it was brought up by all three reviewers. The distinctive pattern led us to be confident that the horizons were the same, and we added to Figure 2 to justify this. We also include a second set of analyses showing that the main conclusions are supported even if these horizons are excluded.

Technical corrections
Line 164, should be "intact"
Thanks, fixed

**Response to Short Comment by Marie Cavitte**

Short Comment of Brief Communication: New radar constraints support presence of ice older than 1.5 Ma at Little Dome C by Lilien et al.

I would like to bring up a few additional points to the RCs submitted.

*We would like to thank Dr. Cavitte for taking the time to post this constructive comment.*

-I find that the description of radar and modeling work done prior to this study is missing several seminal studies:

*Indeed, our reference list is very short to fit within the "brief communication" limits. With this constraint, as a rule we have only referenced the most recent of a series of papers building up a methodology, in agreement with the editor's recommendation.*

1. the Van Liefferinge and Pattyn (2013) study is never mentioned, in particular in the introduction. However, it was their modeling work that identified promising oldest ice sites all over the East Antarctic Ice Sheet that then determined the sites (and so LDC) that should be further surveyed. This should be discussed in the paper, in particular in the introduction.

*We have added a reference to this paper at line 43 of the original text.*

*While modeling shows that LDC is likely to have old ice (e.g. Van Liefferinge and Pattyn, 2013)...*

2. In the discussion of radar depth/age uncertainties, only the Winter et al (2017) study is cited. However, Cavitte et al. (2016) laid a lot of the ground work (and cited by Winter al (2017)). In addition, there is also the work of MacGregor et al (2015) that should cited here.

*It is true that Winter et al. were not the first to quantify radar depth/age uncertainties, and this extends much farther back than MacGregor et al. However, the components are nicely summarized by Winter et al., and we have a tight reference limit, so we have left this as simply ""see Winter et al. (2017) and references therein".*

-Line 39-40: Van Liefferinge et al. (2018) also used the airborne radar data to refine their basal conditions.

*Added that reference.*

-Line 48-49 (introduction) and Section 4.1: a diffuse reflector at depth in the radar data was already observed in the previous radar surveys, both in the airborne OIA data and the GPR DELORES survey data. This was mostly discussed during BEOI meetings (which could perhaps be mentioned as BE-OI pers. comm.). Some of it was included in the publicly available thesis Cavitte et al (2017) available here: https://repositories.lib.utexas.edu/handle/2152/62593

*Thanks, we have noted this at both locations.*

*"In addition to the echo-free zone, some radargrams showed a diffuse horizon near the bed, hinting that basal ice in the area may have different physical properties than the overlying ice (Cavitte, 2017)."*

-Section 4.2: why use such a simple model for dating when there's a 1D inverse model that has shown it works really well in the region (Parrenin et al., 2017 as cited), and Fred Parrenin is one of the co-authors of this paper?

*Our methods differed from Parrenin et al., 2017, in three ways: 1. They used a thermal model and inferred a melt rate and 2. They used a Lliboutry velocity profile while we used a Dansgaard-Johnsen profile. 3. We fit an effective thickness where as they used a measured thickness (though*

Fred has since used the effective thickness as well). We already had used the same pseudo-steady conversion, which is the other distinctive aspect of that work.

Parrenin et al., found no melt in the vicinity of the chosen site, so the first difference becomes moot at BELDC. Since we are looking at a location where there is independent evidence of a lack of basal melt (from prior radar surveys, from lack of draw-down in our radargram, and from Parrenin et al.), it makes sense simply to exclude this possibility from the model. Hence, there is no need to include a thermal module in our modeling.

The remaining substantive difference in our modeling compared to Parrenin et al. was that we used a Dansgaard-Johnsen velocity profile while they used a Lliboutry profile. That choice was made to keep the solution analytic, and justified because the resulting age profiles from the two models are usually very similar. However, we acknowledge that since the more complex model has been applied to this area in the past, it is better to avoid retreating to something simpler. We have now switched to a Lliboutry velocity profile; none of the major results have been changed, and all numbers have been updated accordingly.

The new description of the model is moved in response to other reviews and is now in Sec. 2.3.

-I wonder how the deepest isochrones traced in the manuscript were dated as they seem to get cut off around 30 km from the EDC site.
This concern was raised by the other two reviewers as well. We have made several changes to clarify, noted at the beginning of this document. In short, we used the pattern of the horizons to re-identify them across gaps, but we acknowledge that this has significant uncertainty and now also present results excluding these horizons from the analysis.

-This was not out at the time of submission but there is now a paper (Cavitte et al., 2021) in ESSD Discussions that summarizes all the isochrones that have been traced in the region using the UTIG/OIA/DELORES data sets. It shows that a number of isochrones that are older than 400 ka could be traced in these radar surveys. These deepest isochrones could not be traced to the EDC site due to the steep topography of the Concordia Subglacial Trench, but were dated using the Parrenin et al. (2017) inverse model. The deepest two isochrones in Cavitte et al (2021) show ages of ∼610+/- 35 ka and ∼709+/-54 ka. So, similar ages to those obtained at depth in the manuscript under review. It would be great to see this work referenced and included in the discussion of this manuscript.
We have updated the paper's language to distinguish between ice-core-dated and model-dated reflections, and added a citation to this work. In the most relevant passage, we now write
*Of these, six isochrones were older than 400 ka, the age of the oldest previously isochrone previously dated and traced from EDC. The oldest isochrone that could be continuously traced to BELDC was dated to 465 ka. This increase in the age of dated isochrones introduces a significantly tighter constraint upon the age and age resolution at the LDC site than was previously available.*

It becomes very difficult to make a fair comparison if we mix the two types of dating. Our survey has reflections near LDC that would date to >1.5 Ma with the Parrenin et al. model. We cannot justify such dating in this work, though, since the goal is better constraining a model with the radar, not vice-versa. For a completely fair comparison, we have a truly continuous horizon at 465 ka at EDC, which extends the oldest ice-core-dated horizon in the area by 66 ka, and we have the horizons with distinctive patterns but some discontinuity that extend much older.

---

## Author Response (AR2)

Dear Dr. MacGregor,

Thank you for your continued consideration of this manuscript. We have addressed these technical points in the new version.

- TC convention for the main text (not captions) is to generally abbreviate "Section" as "Sect." (not "Sec.") and "Figure" as "Fig.". See https://www.the-cryosphere.net/submission.html
Fixed. Equation references are corrected to match the style guide as well.
- 56: Please convert to decimal degrees, which are simpler and more modern.
Done.
- 69: 7-12 m/s seems remarkably fast for a ground-based survey of a small region. Could you double-check that these values are accurate?
Indeed, this was in km/h. Thank you for catching the error.
- Figure 1 caption: "BedMachine Antarctica v2" not "BedMachine v2".
Fixed
- In geology, the convention is to use "yr" as the abbreviation for year when referring a period of time and "a" as the abbreviation when referring to a past time. See https://www.geosociety.org/gsatoday/archive/22/2/article/i1052-5173-22-2-28.htm for a discussion. By that IMO sensible convention that I readily admit I have not always adhered to in the past myself, e.g., "41-kyr" cycles should be used rather than "41-ka", etc. Please review the MS with this in mind. In my instances, I believe "kyr" should be used rather than "ka".
Thank you for the reference. Everything is updated to match this convention.

Sincerely,
David Lilien